# The worn-out syndrome: Uncertainties in late working life triggering retirement decisions

Marie Gorm Aabo, Katrine Mølgaard, Aske Juul Lassen * 

Copenhagen Centre for Health Research in the Humanities, Saxo-Institute, University of Copenhagen, Copenhagen, Denmark

☯ These authors contributed equally to this work.
* ajlas@hum-ku.dk

**Data Availability Statement:** The data cannot be shared publicly because we are unable to anonymize data sufficiently to make it publicly available, and interlocutors have signed a statement ensuring that their data will be stored

## Abstract

In recent decades, the extension of individuals' working life has been construed as an important policy issue in Western Europe. Here, retirement causes have been thoroughly researched in a quantitative way, but there is a dearth of qualitative studies on the subject. Through ethnographic fieldwork, we study the complex pathways that lead to the retirement of senior employees in the finance and production industries in Denmark. In particular in the finance industry, we find an insidious uncertainty haunting senior employees regarding their capacity and reputation. We term this uncertainty worn-out syndrome, demonstrating how many interlocutors fear that they are beginning to be seen as worn out, for example, less productive, less motivated, and too old to work. To some extent, this syndrome resembles the impostor syndrome, but it differs in one important aspect: the senior employees are mostly confident about their own skills. Worn-out syndrome is triggered by stereotypes and implicit ageist remarks by colleagues and managers. We show that the worn-out syndrome appears in at least three different ways: as a fear of already being worn out, as a fear of being perceived as worn out by colleagues and managers, and as a fear of becoming worn out in the future without realizing it in time. We argue that current retirement decisions are often fueled by this syndrome and that it leads to abrupt and untimely retirement decisions.

## Introduction

With the modern focus on extending working life and late working life trajectories, the reasons for staying or leaving the job market have been scrutinized by various scholars [1–3]. Often, the decision of retiring or staying has been construed through a push-and-pull framework [4] that differentiates the reasons for leaving the job market, on the one hand, and the reasons for retiring, on the other hand. This framework has been amended with the notions of jump (highlighting the individual decision to retire) [5] and stuck and stay (highlighting the different reasons for continuing working) [6]. Although this latter development has brought new nuances to the field of retirement studies, they have mostly been examined through quantitative surveys. Hence, understanding the complexities and nuances of contemporary retirement decisions remains limited.

securely. Many of the interlocutors come from small towns and have very specific descriptions of highly specialized work in the interviews, which would be practically impossible to anonymize. As such the data contains sensible information. Moreover, all data is in Danish, which would make it difficult for international researchers to access, and this kind of ethnographic data is generally impossible to reproduce. Requests for data access can be sent to the data management officers at datamanagement@hum.ku.dk.

**Funding:** • AJL • 2 grants awarded: 19-0191 and 20-0721 • Velliv Foreningen funded both awards • https://www.vellivforeningen.dk • The funders had no role in study design, data collection and analysis, decision to publish, or preparation of the manuscript.

**Competing interests:** The authors have declared that no competing interests exist.

In the current paper, we examine later working life and the collective and individual negotiations and practices preceding the decision to retire or stay. We show how the retirement decision is complex and often motivated by uncertainty regarding how others see one's working capacity and decline. We do this through an ethnographic data set collected in the finance and production industries in Denmark from 2019 to 2021. We interviewed and followed senior workers, managers, HR staff, shop stewards, and retirees at eight small- or medium-sized companies in the finance and production industries to understand what constitutes a good senior working life. In this process, we have studied the different ways of practicing senior working life and the implicit and subtle ways of negotiating extensions and endings of working lives.

We argue that when studied qualitatively, the borderland between work life and retirement reveals itself as a liminal phase where uncertainty is a constant. The possibilities for managing this uncertainty have been highlighted by others as socially structured and varying based on health, finances, and social background [7]. In our study, we have found uncertainty is more frequently found among white-collar workers in the finance industry than among blue-collar workers in the production industry, and therefore our analysis is based on data from the finance industry.

The financial industry in Denmark includes banks, pension funds, mortgage institutions, fintech companies, asset managers and a number of small businesses. We have focused particularly on banks because this part of the industry has undergone major changes over the past 10 years. In particular, new rules on money laundering following the financial crisis of the 00s, digitalisation and the dismantling of teller services. The finance industry in Denmark has struggled in recent years with the mental health of its employees, and the industry faces major challenges with stress [8]. As such, the uncertainty might be more apparent in the finance industry than in other service sectors, but we do not have data to show this. In 2021, there were 26 banks in Denmark with a total of around 35,000 employees. In this research project, we focus on small and medium-sized banks. We interviewed workers from several different areas, including private advisors, back-end employees, business advisors and directors.

In numerous interviews, we have found our interlocutors in doubt about how they are perceived by their peers at the workplace, leading them to ask questions such as the following: Is it time to retire? Can I keep up with the speed at the workplace? Would my manager or colleagues even tell me if they thought it was about time? Have I damaged my status and role by staying so long? Do they see me as worn out?

Based on this uncertainty, we suggest that to a considerable degree, current retirement decisions are fueled by what we term *the worn-out syndrome*. The worn-out syndrome is the feeling that it might be time to retire because the individual has a hunch that their managers and coworkers might think that they are beginning to be too slow, forgetful, and so forth.

In this sense, the syndrome relates to the concept *age metastereotypes* i.e. the "*stereotypes that older (. . .) workers believe that others hold about their group.*" [9, p. 26] We elaborate on one particular proposition that Finkelstein et al. suggest in order to examine the complexities of the ways in which age metastereotypes shape interactions at work, i.e. (proposition 8): "*Negative age metastereotypes may lead to threat reactions, including emotions such as fear, worry, and/or sadness, and anticipation of the possibility of conforming the metastereotype.*" [9, p. 35]. Moreover, worn-out syndrome relates to McGonagle and colleagues' concept of *perceived work ability*, which they define as "*. . .an individual's self-perception or evaluation of his or her ability to continue working. . .*" [10, p. 3]. Personal factors such as health and sense of control play a crucial role in one's perception of work ability and therefore affect decisions of withdrawal or retirement. We find that such factors are influenced by stereotypical notions of older workers, which are more or less unconsciously shared in the workplaces. As such, our study seeks to elaborate on the complex experiences related to metastereotypes and perceived work ability.

Much like impostor syndrome [11] even though it occurs at the opposite end of working life, worn-out syndrome stems from a feeling of inadequacy that is seldom shared with coworkers. Because worn-out syndrome is most apparent in the finance industry, we mainly use data from this part of the fieldwork, though our results are developed with data from both the finance and production industries.

We start our analysis with an ethnographic vignette before displaying the different ways in which this syndrome plays out in practice. However, before we dive into this, we clarify and conceptualize worn-out syndrome and then briefly contextualize our argument by relating it to the Danish labor market, as well as current research streams regarding the role of age in the labor market. This contextualization is followed by a note on the design and methods applied in the present study.

## The syndrome

With worn-out syndrome, we attempt to conceptualize and broaden the comprehension of a feeling that we have continuously encountered in our fieldwork: the feeling that it might be time to retire because of a fear of no longer being able to keep up with workplace expectations. As already mentioned, worn-out syndrome shares some similarities with impostor syndrome in the sense that both terms describe a feeling of inadequacy in the workplace.

Impostor syndrome was first mentioned in Clance and Imes' 1978 article "The Impostor Phenomenon in High Achieving Women: Dynamics and Therapeutic Intervention." As the name of the article suggests, the concept is linked to highly successful women, and it describes an "*internal experience of intellectual phoniness*" [11, p. 241]. Even though the studied women had achieved a lot of recognition and success at their work and school, they did not feel that the recognition was in line with the image they had of themselves. Thus, the concept identifies the belief that one's success is not justified and that one has fooled others into believing that they are as intelligent as they seem to be. Earned degrees, professional recognition, and other acknowledgments that are often seen as evidence of success are devalued by those who suffer from impostor syndrome; instead, these accolades are perceived as matters of luck, temporary effort, or misjudgments from professors. Since the publication of the article, the concept has been viewed as a phenomenon that both men and women suffer from when they doubt their abilities and accomplishments and feel like a fraud. Moreover, it has been criticized for individualizing the causes of the syndrome and neglecting its structural foundations [12].

To sum up, impostor syndrome describes an internal belief that one is not skilled enough, even though the person has achieved recognition at work. Thus, there is a discrepancy between the inner feeling of not being good enough and the outer recognition of one's achievements and abilities at work.

We argue that worn-out syndrome appears in at least three different ways: First, in a few instances in our data, we see that senior employees feel that they are not able to keep up with the expectations at work. Here, worn-out syndrome resembles impostor syndrome in the sense that one feels inadequacy. Second, the senior employees are often afraid of the ways their colleagues and managers might perceive them as being worn out. One could argue that this fear of what others might truly think of you is also at stake in impostor syndrome. However, we have seen that most of our interlocutors are actually confident in their own abilities at work. In this sense, worn-out syndrome differs from the impostor syndrome because it does not necessarily stem from an experienced inadequacy but from an imagined external view of oneself. Third, senior employees are often afraid of future decline. They dread becoming worn out without noticing it themselves. Here, worn-out syndrome also differs from impostor syndrome because it relates to a potential future inadequacy.

## The Danish senior labor market

Denmark is a welfare state with public universal pensions that are currently attainable from 67 years of age (66 years of age during data collection). Based on the 2004 "welfare reform," the statutory retirement age has increased from 65 years of age in recent years and will continue to increase in the coming decades. As expected, this has led to a rapid increase in labor market participation for those over 65 years old. There are only a few sectors with an obligatory retirement age, and as such, the retirement timing and decision is individualized and left to a negotiation between the employer and employee; however, in most cases, employees retire at the statutory retirement age. While economy is a factor in retirement decisions, the pension system ensures a sound economy for a large part of the Danish retirees. As such, staying or leaving the work force might be more determined by factors such as the worn-out syndrome in Denmark than in other countries, where economy is a larger driver for continuing to work.

The Danish labor market is organized around collective agreements. Every three years, the unions and employer associations from each industry negotiate the working conditions (wages, hours, etc.). In recent years, senior rights have become increasingly widespread in collective agreements, and often, the senior employees (often from 58 years of age) have the right to a weekly day off. As part of the focus on senior working life, many workplaces have organized annual interviews—so-called senior interviews—with senior employees to engage in a dialogue about plans regarding continuing or retiring. Although this initiative would seem an ideal place to prevent and discuss ideas and feelings of being worn out, we will later show how this is seldom the case: senior employees are careful not to show signs of decline and plans of retiring at such interviews.

## The role of age and stereotypes in the labor market

Worn-out syndrome relates to a range of stereotypes surrounding senior employees. A wealth of studies have demonstrated that senior employees are the targets of stereotypes and prejudices [13]. For example, senior employees are often seen as less productive, less motivated, more resistant to change, and having a lower ability to learn. Moreover, studies have shown that senior employees also hold prejudices about their own age group [14, 15]. Some studies indicate that age stereotypes influence the ways seniors judge themselves [16, 17] and that early retirement intentions are often linked to the negative stereotyping of senior employees [18]. Such negative stereotyping and ageism is part of the language at some workplaces at times restricting senior workers from participating in training, and which can lead to a widespread 'too old for' narrative [19, 20]. With the concept of worn out syndrome we elaborate on the findings of embodied ageism and 'too old for' narratives. However, we wish to focus on the elements of fear and concern relating to such narratives, indicating that the negative stereotypes not necessarily have to be internalized and accepted as a truth in order to play a crucial part in the senior working life and retirement decisions. The participants do not necessarily deem themselves 'too old', but fear that their surroundings do.

However, studies also show that these stereotypes are based on false assumptions and that there is little evidence that job performance declines with age [21, 22]. However, these stereotypes are likely to influence how senior employees' regard their worth and contribution in the workplace because managers, senior employees, and colleagues rely on these false assumptions in their day-to-day decision making [14].

We argue that such false assumptions fuel worn-out syndrome by enforcing a specific way of imagining the life course and blindly assuming that senior workers are declining in their abilities to perform at work. Although decline forms part of the life course for many individuals, there is little evidence that this decline influences the ability to handle one's core tasks at

work [23]. Moreover, senior employees can often compensate for the loss of specific abilities through wisdom and expertise [24]; thus, it could be argued that the labor market should show more acceptance toward signs of decline.

In the past few years, contemporary retirement pathways have been scrutinized [25]. Overall, the retirement trajectories are diversifying, and many senior employees engage in gradual retirement, bridge employment, entrepreneurship, and unretirement. With the removal of mandatory retirement in most countries and industries, the retirement decision has become increasingly individualized [26], leading to a management void. Often, senior employees and managers are afraid to engage in a dialogue about the retirement process, and subtle hints and implicit negotiations leave both parties wondering about the intentions of the other. This can lead to sudden and abrupt retirement decisions; hence, the experiences and competencies of senior employees are consequently disappearing from the workplace. Likewise, the fact that there is no obligatory retirement date can be a problem for managers because some senior employees extend their working life while managers wish to get rid of them and have no legal measures to do so. In some instances, senior employees are actually seen as "worn out" by their managers and colleagues, but as we will show, worn-out syndrome usually occurs independently of this judgment.

## Design and method

The data for the current paper stems from qualitative fieldwork conducted at small- and medium-sized companies in the finance and production industries in Denmark between December 2019 and mid-March 2020 and between April and October 2021. We conducted semistructured interviews (N = 92) with senior workers, their managers, shop stewards, and HR personnel at eight different companies (5 companies in the finance industry and 3 companies in the production industry). We asked them about their senior working life, the senior politics at the workplace, the community at the workplace, and ideas about the future (see interview guide in S1 File). All interlocutors were informed by the purposes of the study and signed a written informed consent declaring we could use data for scientific and dissemination purposes once properly anonymised. The fieldwork was approved by the data management board at the University of Copenhagen. We followed ethical guidelines for ethnographic fieldwork and have ensured that sensible information shared by interlocutors during interviews and participant observations were not shared with their colleagues or managers.

The interviews were conducted by the authors as well as the research assistants XXX (blinded for review). We did not predetermine when an employee was categorised as senior, but allowed the company to set the age limit, when we contacted them. Often in the collective agreements, the workers achieve senior rights from 58 or 60 years of age, but the senior label is often resisted by the employees. The youngest senior worker we interviewed was 58 and the oldest was 70 years old. We interviewed 44 senior workers (22 employed in the finance industry, 22 employed in the production industry), 8 retirees (5 formerly employed in the finance industry, 3 formerly employed in the production industry), 2 shop stewards (both employed in the finance industry), 8 HR employees (6 from the finance industry, 2 from the production industry) and 23 managers (17 employed in the finance industry, 10 employed in the production industry), as well as 7 representatives from unions and pension funds. We did this to study not only the senior employees and their practises at work, but also to understand the culture at the workplace, how their colleagues and managers perceived them and to get a broader understanding of the industries at large. To get to know both the thoughts and practices of the employees soon to retire and the actual actions of people who had already retired and participated in a senior program, we interviewed people still working as well as some who

had retired. We have not interviewed younger co-workers about their views on the senior workers, but as we show, the managers, shop stewards and HR employees gave us an idea of how the skills and experiences of the senior workers were perceived by others.

We also conducted participant observations (N = 25). We were part of the workplaces and participated in practical chores, chit-chat during lunch and coffee breaks, and meetings and workshops. In this regard, to explore the everyday life, values, practices, and ideas as they unfolded in the workspace, we participated moderately according to Spradley's [27] degrees of participation.

Each interview has been transcribed ad verbatim and subsequently coded using NVivo by the authors, research assistants and student assistants. The authors conducted four analytic workshops during spring and summer 2020 guided by the use of constant comparison and inductive coding [28]. With this, we identified themes and ways of articulating the reasons behind working in old age and/or retiring. We used analytic induction [29], as we continuously developed hypotheses in the material and tested these using NVivo codes and collective, close readings of selected interviews. Unless noted otherwise, the quotes are indicative of widely expressed views.

We included companies based on our knowledge of the field. We contacted companies that either had a written senior policy or had an articulated way of managing older workers. Some of the companies we contacted had been nominees for a 2008 and/or 2009 prize called "the senior workspace of the year." Thus, our material and analyses are not representative of small- and medium-sized companies, nor are they representative for the two industries. Rather, we conducted purposive sampling [30] to study the companies that already have senior policies and/or are aware of this management area. We contacted the companies via phone and mail and ended up including five banks and three production companies. In the following, we delve into the story of one interlocutor, Susanne, to show how worn-out syndrome was part of her retirement decision. This is a significant example, chosen because it shows how the syndrome can affect many aspects of senior working life.

## Ethnographic vignette: Susanne

Susanne is 67 years old and has been retired for six months. She is married and has two daughters and three grandchildren. She finds it difficult to identify one specific reason for retiring. Her family situation, a fear of becoming or being worn out, and her development in the industry and changing assignments all seem to have played a part. Educated as a bank assistant, Susanne has held several management positions throughout her working life in the financial sector. She is very confident in her working skills and has never considered fewer weekly working hours or gradual retirement because she does not find it feasible in her position.

> *She* [my former manager] *asked me if I wanted to work shorter hours. But I didn't want to do that because there was no one else to take over if I wasn't there. And all that work—the work I handled by myself—six persons are handling it now.*

(Susanne, 67, retiree, finance)

This recognition means a lot to Susanne, and several times in the interview, she points out how it has been important for her to leave her job with a good reputation while she is still going strong. The last few years before retirement, Susanne was responsible for ensuring that the bank complied with the new money laundering legislation. Susanne refers to this position and the structural changes in the industry as one of the reasons that influenced her decision to retire. With the bank being urged to allocate more resources to money laundering measures,

Susanne saw this as an opportunity to take a step back and train a new employee so that she could retire in the near future. However, things kept getting in the way of her retiring, such as numerous staff turnovers: "*It took a year and a half from the time I told them that I wanted to retire before I got to do it* [. . .] *Things kept coming up that made it impossible for me to say good-bye. With a clear conscience, right*?" (Susanne, 67, retiree, finance).

She did not want to leave before everything was in order. Moreover, one of the reasons for her planning retirement was her daughter's expected pregnancy: "*I said* [to my manager] *'When my daughter gets pregnant, then you have nine months until I retire!'*" (Susanne, 67, retiree, finance). However, she gave nine months' notice three times, but every time her daughter's pregnancy was involuntarily terminated, her reason for retiring disappeared. She found it difficult to retire for her own sake, so her ill-fated family situation also prolonged her working life.

Meanwhile, the new employee she was mentoring also made her contemplate her own position in the workplace. She experienced that her colleagues listened more to him—male, younger, with a military education—than to her. He said the same things she had been saying for years, but now they "*actually listened*." This made her feel uncertain about how her colleagues perceived her. Did they think she was too old? Was she damaging her future reputation by staying too long?

Although Susanne was very confident in her work, she also had a lumbering feeling of inadequacy. She agreed with her manager, Steen, that he should promise to tell her if she was starting to lag behind because she did not want to be the kind of person coworkers are talking about: "*Oh no, why isn't she leaving soon*?" (Susanne, 67, retiree, finance). Susanne has experienced this with a former colleague who stands out as a dreadful way of retiring to her:

> *I've had an employee in my department, where everybody thought the same thing: "Mogens, please stop working now. You have to retire now." But he didn't. He continued working for too long. And when he left, I thought to myself, "Phew, that was about time!" And then we had to sort out the mess he made the past six months.*

(Susanne, 67, retiree, finance).

Throughout the interview, it seems clear that this fear of repeating Mogens' ill-timed exit is a big concern for Susanne:

> *What if you don't realize it? He didn't realize it himself. And he wasn't stupid or anything; he just didn't realize it. Then I said to Steen, "Now, you have to tell me. If you notice that I'm falling behind or if you think that I'm worn out, you will let me know, right?" Yes, he would, he said. But he never told me anything, so there was probably nothing to it.*

(Susanne, 67, retiree, finance)

Although there was probably nothing to it, Susanne retired before she would ever know.

## Results

### Three manifestations of worn-out syndrome

When interviewing Susanne, it is difficult to pinpoint whether her uncertainty and the deal she made with her manager relate merely to future uncertainty and how her colleagues might perceive her or if she felt that she could not perform in her job. "(T)*here was probably nothing to it*," she mentions, which illustrates this ambiguity. Even though Susanne is proud of her career

and is largely confident that her work is satisfactory and appreciated, she is also a bit afraid that it is not. Mostly, she talks about the deal as a type of future insurance, where her manager would have told her if she was worn out. At other times, it seems that she dreads it might already have been past her time. She does not think that her former colleagues saw her in the same way as she sees Mogens, but she cannot be certain.

In this regard, Susanne's worn-out syndrome brings in a prospective future decline, how her colleagues might perceive her as worn out, and if she is actually already worn out when retiring. At times, there might only be one of these aspects at play; at others, these aspects intertwine and are difficult to differentiate. In the following sections, we have separated the three manifestations of the syndrome to analytically unfold the worries and uncertainties related to this syndrome.

### Manifestation 1: Uncertainty regarding one's present performance

Like Susanne, many of our interlocutors who express uncertainty about their status at work are mostly afraid of becoming worn out in the future or being perceived as worn out by their peers. Worn-out syndrome mostly regards other spheres—futures or peers—but not one's current status. However, before retiring, Susanne was also concerned with actually being worn out: *"I was 67 when I left. And the fear of not being able to keep up—I think that was the worst part"* (Susanne, 67, retiree, finance).

Susanne did not want to become like Mogens, who did not know when to leave. Although Susanne talks about it as a concern for the future, it seems that she was also concerned about her present status when she retired. One might say that this uncertainty regarding not being able to notice one's own decline relates just as much to the present as to the future. If one cannot tell in the future, how can one tell now? As such, worn-out syndrome also concerns one's present decline and job performance.

With the other interlocutors, worn-out syndrome concerns present job performance in more immediate ways. Leif, a 65-year-old senior employee at a bank, is considering retiring when we interview him. His reasons for retiring are manifold and involve his partner's chronic disease, but he also believes that his mission in the bank has come to an end. He has driven their pension department for years, and now, new ways of banking mean that they have to learn new methods in the small department, which consists of three persons. He has to either engage in new courses or become outdated and obsolete. He has an idea that he could learn the new system, but a part of him also doubts he could:

> *Now is the time* [to retire]. *I would have had to spend so much time and energy learning this new system. I could have done it, I think, if I wanted to. But I don't want to.*
>
> (Leif, 63, senior employee, finance)

The effort he would need to invest to stay up to date was too large for him. Part of his doubts also include the IT skills of his younger colleagues. It would take him more time to learn the new system than them, and he is not feeling at par with his younger colleagues in this aspect. Usually, this would not be a problem in everyday assignments, but when learning a new system and working method, this could become an issue. As such, Leif's doubt and retirement decision does not concern his job performance, rather the idea that he will need to invest more than before. He is a bit uncertain whether his refusal to do so stems from a lack of will or a fear of not being able to.

One of Leif's assignments is to have meetings with clients who want to discuss their pension savings. The clients' personal advisers would book him or one of his colleagues from the

pension team for the meeting. Since Leif announced his retirement, many of the clients' personal advisers stopped booking him when they arranged the pension meetings. Although Leif, on the one hand, thinks this was caused by the fact that he would soon disappear—and as such would not be able to have follow-up meetings with clients—on the other hand, he is concerned that it could be caused by him not being able to do his job adequately.

Leif's decision to retire is also influenced by his recognition that he did not give much effort to his job performance in recent years. Again, he is uncertain whether this was caused by a lack of energy—and as such could be ascribed to his aging process—or by a lack of motivation, in which case he would not ascribe it to aging. Nevertheless, he recognizes that he would need to put in more effort if he should continue working. To others, this manifestation of worn-out syndrome related to current job performance has opposite consequences.

Bent, a 63-year-old senior employee at a bank, feels that he needs to put in extra hours because of his uncertainty regarding his performance. He experiences that he is getting slower at his assignments and fears that his manager and colleagues might notice this. Instead of talking to his manager or peers, his strategy is to hide his decreasing pace by putting in extra hours without registering them. As such, his senior work life is busier than ever.

In Leif's, Bent's, and Susanne's cases, worn-out syndrome concerns an uncertainty regarding whether they perform adequately in their present job. For Susanne, this is expressed as a concern for being ignorant of the hypothetical decline that might already have happened, while Leif and Bent can point to specific assignments and job functions that they subjectively feel unable or unwilling to perform adequately, even though their managers think otherwise. However, as we have shown, their uncertainty regarding their current job performance is mixed with a concern for how colleagues perceive them and how they might become worn out in the future. These two manifestations of worn-out syndrome are more common in our data than uncertainty regarding their current performance and are the foci of the following two sections.

## Manifestation 2: Uncertainty regarding colleagues' and/or managers' perceptions

Worn-out syndrome also manifests through senior employees fearing that their colleagues and managers perceive them as being worn out. Even though most of our interlocutors are confident in their abilities and are aware that they often possess a special skillset and a great deal of highly specific knowledge and experience, this uncertainty is common in our data. Senior employees, their colleagues, and their managers express a range of stereotypical assumptions about older people in general that seems to fuel this manifestation of worn-out syndrome. When older people in general are talked about as slow or unproductive in the workplace, the interlocutors notice this, though their colleagues do not find that these characteristics to apply to them.

Sometimes, the uncertainty is triggered by a specific experience, while at other times, it is more subtle—like a gut feeling that something has changed in the way one's colleagues or managers perceive oneself or simply just a fear of being perceived as worn out, even though there are no indications that this should be the case.

Leif, the 65-year-old head of pensions, refers to a specific experience triggering his uncertainty. As we sit down to talk with him, he is beginning to feel that his colleagues are perceiving him differently from earlier. He has one specific moment when he felt this for the first time:

*We went on a company outing a year and a half ago, and somebody came up with the remark, "Well, aren't you retiring soon?" [. . .] In that moment, I got a little pissed off, to put it bluntly.*

*It wasn't fair for him to ask me that.* [. . .] *It's the feeling that you're not good enough any-more. What you do and the way you behave is not sufficient. . ..*

(Leif, 63, senior employee, finance)

Even though Leif's colleagues see him as a highly valued and skilled worker, this comment made him evaluate his future career; it made him doubt whether he was still capable of doing the things he used to do. It also made him aware that he sometimes needed to work harder and longer to finish the same tasks as his colleagues and to keep up with his own expectations regarding the quality of his work.

Leif is not alone with this feeling of being perceived differently than before because of his age. As seen with Susanne, she found that her colleagues listened more to her new colleague, whom she was mentoring, than to her both her and her mentee explained the same matter. Likewise, Leif is not the only one who works extra hard to keep up with his own expectations. Bent, who works overtime without telling anyone, states that he is nervous about making minor mistakes. He highlights an incident where he got the name of an important client a little wrong, and one of his managers noticed it and pointed out the mistake to him:

*Maybe I was a bit more annoyed by or nervous about making that mistake. Because I made it as a 62-year-old and not as a 42-year-old. I feel a bit more insecure now; I do.*

(Bent, 63, senior employee, finance)

Bent was afraid that his managers and colleagues would interpret his mistakes as a sign of his age. He talked with his managers at the annual performance and development review about the incidence, and they were trying to assure him not to worry about it. However, Bent cannot put away the uncertainty regarding how they perceive him. To counteract this uncertainty, he doublechecks and works overtime—usually without telling anyone:

*I think I doublecheck all of my e-mails, my texts, my credit setting, and my documents one or two times extra. Because of my age. Not because that, I think that I'm losing my touch. I just want to be completely sure that nobody else will get the idea that I'm losing my touch.*

(Bent, 63, senior employee, finance)

Bent is afraid of making mistakes and being too slow at completing his tasks, and because of that, he takes longer to complete his work, which means he needs to work overtime. Bent's fear of whether his forgetfulness is caused by aging—and whether his colleagues might think it is caused by aging—resembles the study by Erber and colleagues [31] showing that memory failures of young and older adults are judged differently. If an older person had memory failure, the participants were more likely to ascribe such failures to aging and mental difficulties.

In other instances, we find senior workers who had a gut feeling that their colleagues and managers saw them as worn out. Our interlocutors are afraid that even though no one would say anything out in the open or directly to them, their managers and colleagues would talk negatively about them behind their backs. Most of the interlocutors would not dare to discuss how they were perceived with colleagues and managers. Others have discussed it openly with the manager during the annual performance and development review. Thus, some of the managers we interviewed were aware of the issue. Finn, a 62-year-old manager at a bank, explains that he found it important to recognize senior employees and let them know that they are valued:

*If nothing else, you have to motivate them a little more* [so they will think to themselves]: *Well, okay, I may feel that I'm about to stagnate a bit, but my bosses say that I matter to the team and that they still really want me working here, so let's just continue for one more season*!

(Finn, 62, manager, finance)

Even though showing appreciation and praising the older employees helps, Finn's senior employees are still uncertain about how they are perceived. Ensuring them that they are not worn out does not completely cure this syndrome, as will be evident in one of Finn's employees, Niels, in the coming section. Rather, the uncertainty surrounding late working life seems to be a constant that is deeply embedded in the ways in which the aging process is perceived and fueled by stereotypes.

## Manifestation 3: Uncertainty regarding future status

The third way that we find the syndrome manifests is the fear of future decline. Often, this fear relates both to how the individual's colleagues might perceive them in the future as well as how the individual might not be able to tell whether they are worn out. As we saw in the previous section, former colleagues who ill-timed their retirement become bad examples to avoid. Susanne's fear of ending up like Mogens is one of the reasons she retired, even though she feels quite a lot of pride and confidence in her own abilities. Moreover, as we saw, Susanne used her manager as part of her strategy to avoid becoming like Mogens by agreeing to tell her if it was time to retire. In Susanne's case, the fear of future decline is much related to a fear of losing one's self-knowledge and situational awareness with age and, therefore, not being able to decide for oneself when it is the right time to retire.

In our data, the fear of losing one's situational awareness is a stereotypical way of depicting older people and retirees. Susanne talks about former, retired colleagues who continued to visit the workplace when she worked there. Then, they would come and chit-chat for half an hour and disturb Susanne in the middle of her work. Susanne interprets this as a lack of situational awareness in her former colleagues. Hence, after retirement, Susanne does not show up at the bank to talk. She does not want to be disturbing anyone.

This fear of future decline has an impact on many of our interlocutor's retirement decisions. Børge, a 65-year-old retiree and former employee at a bank, explains, like Susanne, that one of the reasons he retired was the fear of a future decline that he would not be aware of:

*I didn't want for anybody to be the person who had to pull me aside and ask me, "You know what, isn't it time for you to go on a very long vacation? Don't you need to go home and be there for your grandchildren?"* [laughs].

(Børge, 65, retiree, finance)

Even though Børge recognizes that he had not "as much coffee left in the jug," which is how he describes a decrease in energy, he is confident that he was doing an adequate job for the last few years of his working life. The point then, for Børge, was to quit while ahead rather than to risk ending up in a place where one can no longer keep up. Much like Susanne, Børge was afraid of losing the ability to be aware of his stage in the aging process and, therefore, that he would be unable to retire in time. As he puts it, "*It's better to stop early enough for people to ask you 'why?' instead of 'when?'*" (Børge, 65, retiree, finance). If they ask "when," this implies it should have happened a long time ago. Often, as we see with Susanne, interlocutors mention bad examples of previous colleagues who retired too late.

Following this logic, retirement should be timed prior to any detectable signs of decline. Niels, a 64-year-old senior employee at a bank, also talks about the uncertainty of losing the ability to determine when it is the adequate time to retire:

Niels: *There will be a limit at some point.*

Interviewer: *And would you then think by yourself: Now is the time to stop working?*

Niels: *Yes, if it's me who realizes it, right? It might be my surroundings realizing it at first: "It's about time, Niels, you are here* [at the workplace], *but you are not even supposed to be working today!"* [Laughs].

(Niels, 64, senior employee, finance)

Although Niels jokes about it and states that he is not currently worried, he foresees that this concern will be part of his retirement considerations. Decline will inevitably come. The question is whether he can tell when it does.

In the cases of Susanne, Børge, and Niels, we have seen how the fear of future decline does not merely relate to a fear of decay, loss of skills, and old age. It also relates to the fear of losing one's self-knowledge and situational awareness. Although these three interlocutors are all generally confident about their own skills—and although all three are highly skilled workers who are much valued by their colleagues and managers—an uncertainty about their present state is part of their retirement considerations. They might joke about it and mostly refer to uncertainty as a future state, but the uncertainty seems to appear insidiously.

As numerous studies have shown, deciding when and how to retire is critical to identity and postretirement well-being [32–34]. The ideal is an autonomous and well-planned decision taken at the right time before anybody can doubt one's skills, energy level, and cognitive functions.

## Discussion

Worn-out syndrome seems to be scaffolded by a range of stereotypes about older workers and their abilities. As others have shown, older workers are targets for implicit kinds of discrimination [35, 36], and they are often seen as less motivated, less flexible, less healthy, and so forth [37]. Although the managers and colleagues in our study repeat many of these stereotypical ways of seeing older workers, they tend to state that this is not the case with their senior employees or colleagues. Despite this, it seems that these stereotypes influence how the senior employees perceive themselves and their status in the workplace. As we have shown, such stereotypes are often implicitly present when Leif is asked when he will retire or when Susanne's coworkers listen more to her inexperienced mentee than to her. And as we have shown, worn-out syndrome can occur when such stereotypes appear in implicit ways–e.g. when Susanne experiences her colleagues listening to her younger colleague instead of her, or when Leif is asked whether he retires soon–but also when Bent experiences he needs to put in extra hours to keep up with the workload, or when Leif questions whether he has the ability to learn a new system. In such cases, Bent and Leif might actually be worn out, but they cannot know, as the mere fear of being worn out impedes them from discussing whether their problems are caused by their own inability to keep up or by demands that they nor their younger peers are able to keep up with.

Although the stereotypes portrayed above inform the working environment of the senior employees and conditions of worn-out syndrome, they also reveal how a particular idea of the life course and the aging process itself can guide late working lives. As shown by (blinded for

review), the stairway of life—on which we ascend in the first half of life, peak and then descend to old age—forms our way of thinking about older persons and the ways we embody the aging process. In this idea of the life course, decline starts as soon as we have peaked in the middle of life. This holds importance regarding late working life because it seems that the contemporary labor market leaves no space for decline. When senior employees have the slightest hunch that they are worn out, that their coworkers see them as worn out, or that they might soon become worn out, retirement seems to be the most obvious solution. They retire, preferably before anyone has noticed their decline.

In recent decades, gradual retirement programs have been portrayed as a way to prolong working lives [2], but such programs have gained less uptake than expected in most countries. One reason for this could be the perceived signs of decline that engaging in a gradual retirement scheme would imply. In our study, most senior employees seem eager to comply with the norms of a full-time working week because working part-time could be seen as a sign of decline.

As Lamb has shown [38], in Western societies, the ideals of successful aging have led to a stigmatization and individualization of decline instead of a meaningful and accepted part of the life course. Although Lamb's studies demonstrate that this idea conditions intergenerational relationships and how we live in old age, our study shows that the contemporary labor market has no tolerance for decline. Although this is informed by ideals of productivity, it also seems to be informed by a fear of decline in general, even though senior employees could compensate for a hypothetical decline through their expertise.

Although we have conducted as much fieldwork in the production industry as in the finance industry, all the examples in the present article are from the financial industry. The fear of decline is more apparent in the financial industry than in the production industry. This does not imply that ideas of decline and decreasing productivity are completely absent in the production industry. One reason for this could be that whereas being worn out in the finance industry has mostly appeared as cognitive decline, in the production industry, being worn out mostly appears as physical decline. This difference between different kinds of decline has been problematized and politicized in different ways in the public debate. Whereas physical tear and tough working conditions in the production industry have been a subject of political debate and union fights for more than a century, cognitive decline and mental health are a more recent issue of discussion. The worn-out body in the production industry is construed as a collective and political matter of concern, whereas the worn-out mind is often portrayed as an individual problem. In the finance industry, cognitive decline is an implicit and hidden fear. When senior employees find no way of talking openly about their fear and uncertainty regarding their decline, the risk of experiencing worn-out syndrome is more apparent. As such, to counteract worn-out syndrome among white-collar workers, cognitive decline and mental health need to be politicized to a larger extent.

## Conclusion

Throughout the present paper, we have demonstrated that senior employees—in particular in the finance industry—are uncertain about their status in the workplace. Mostly, the origin of this uncertainty is difficult to pinpoint but relates to stereotypes regarding older workers, situations where the senior employees (mis)interpret signals from their coworkers, past examples of coworkers retiring too late, a lack of self-confidence when comparing work results with younger colleagues, and a lack of tolerance regarding any signs of decline in the labor market. Often, worn-out syndrome forms part of the retirement decision.

We have identified three different ways in which the worn-out syndrome manifests: (1) a fear of an already occurring decline, (2) a fear of being perceived by colleagues as declining,

and (3) a fear of future decline. However, senior employees and their skills are often highly appreciated by colleagues and managers, and most of our interlocutors actually think of themselves as capable and skilled. This contrast between uncertainty and appreciation is largely caused by the stereotypes of older workers as vulnerable and in decline. Thus, we suggest that stereotypes and ageism targeting older workers play a significant part in triggering worn-out syndrome. Often, the syndrome does not stem from what peers think about the senior employee but rather from what the senior employee thinks the peers think about the senior employee—or what they might think in the future.

Although we have conducted fieldwork in both the finance and production industries, worn-out syndrome is much more apparent in the finance industry. All the interlocutors and quotes in the current paper stem from this area of the fieldwork. Although worn-out syndrome might also be at play in the production industry, it is so to a much lesser degree, which could be explored in a different article. Thus, it could be argued that this syndrome mostly occurs in the finance industry. What we would suggest instead is that worn-out syndrome forms part of the retirement decision in other industries, in particular in those with a high level of mentally challenging assignments.

## Supporting information

**S1 File. Interview guide for senior staff.**
(DOCX)

## Acknowledgments

The authors would like to thank the participating work places and in particular their employees participating in the study. Moreover, we would like to thank the director of the Copenhagen Centre for Health Research in the Humanities, Astrid Pernille Jespersen.

## Author Contributions

**Conceptualization:** Marie Gorm Aabo, Katrine Mølgaard, Aske Juul Lassen.

**Data curation:** Marie Gorm Aabo, Aske Juul Lassen.

**Formal analysis:** Marie Gorm Aabo, Aske Juul Lassen.

**Funding acquisition:** Aske Juul Lassen.

**Investigation:** Marie Gorm Aabo.

**Methodology:** Marie Gorm Aabo, Aske Juul Lassen.

**Project administration:** Marie Gorm Aabo, Aske Juul Lassen.

**Software:** Marie Gorm Aabo.

**Writing – original draft:** Marie Gorm Aabo, Katrine Mølgaard, Aske Juul Lassen.

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
