## [Decision Letter · Decision Letter 0]

24 Aug 2022

PONE-D-22-20359The worn-out syndrome: Uncertainties in late working life triggering retirement decisionsPLOS ONE

Dear Dr. Lassen,

Thank you for submitting your manuscript to PLOS ONE. After careful consideration, we feel that it has merit but does not fully meet PLOS ONE’s publication criteria as it currently stands. Therefore, we invite you to submit a revised version of the manuscript that addresses the points raised during the review process.

We look forward to receiving your revised manuscript.

Kind regards,

Daphne Nicolitsas

Academic Editor

PLOS ONE

Journal Requirements:

When submitting your revision, we need you to address these additional requirements. 1. Please ensure that your manuscript meets PLOS ONE's style requirements, including those for file naming. The PLOS ONE style templates can be found at https://journals.plos.org/plosone/s/file?id=wjVg/PLOSOne_formatting_sample_main_body.pdf and https://journals.plos.org/plosone/s/file?id=ba62/PLOSOne_formatting_sample_title_authors_affiliations.pdf 2. We note that the grant information you provided in the ‘Funding Information’ and ‘Financial Disclosure’ sections do not match. When you resubmit, please ensure that you provide the correct grant numbers for the awards you received for your study in the ‘Funding Information’ section. 3. Thank you for stating the following in the Acknowledgments Section of your manuscript:  "Moreover, we would like to thank Velliv Foreningen, who supported the617 study, as well as the director of the Copenhagen Centre for Health Research in the Humanities,618 Astrid Pernille Jespersen." We note that you have provided funding information that is not currently declared in your Funding Statement. However, funding information should not appear in the Acknowledgments section or other areas of your manuscript. We will only publish funding information present in the Funding Statement section of the online submission form.  Please remove any funding-related text from the manuscript and let us know how you would like to update your Funding Statement. Currently, your Funding Statement reads as follows:  "•
AJL •
2 grants awarded •
Velliv Foreningen funded both awards •
https://www.vellivforeningen.dk •
The funders had no role in study design, data collection and analysis, decision to publish, or preparation of the manuscript" Please include your amended statements within your cover letter; we will change the online submission form on your behalf. 4. We note that you have indicated that data from this study are available upon request. PLOS only allows data to be available upon request if there are legal or ethical restrictions on sharing data publicly. For more information on unacceptable data access restrictions, please see http://journals.plos.org/plosone/s/data-availability#loc-unacceptable-data-access-restrictions.  In your revised cover letter, please address the following prompts: a) If there are ethical or legal restrictions on sharing a de-identified data set, please explain them in detail (e.g., data contain potentially sensitive information, data are owned by a third-party organization, etc.) and who has imposed them (e.g., an ethics committee). Please also provide contact information for a data access committee, ethics committee, or other institutional body to which data requests may be sent. b) If there are no restrictions, please upload the minimal anonymized data set necessary to replicate your study findings as either Supporting Information files or to a stable, public repository and provide us with the relevant URLs, DOIs, or accession numbers. For a list of acceptable repositories, please see http://journals.plos.org/plosone/s/data-availability#loc-recommended-repositories. We will update your Data Availability statement on your behalf to reflect the information you provide. 5. PLOS requires an ORCID iD for the corresponding author in Editorial Manager on papers submitted after December 6th, 2016. Please ensure that you have an ORCID iD and that it is validated in Editorial Manager. To do this, go to ‘Update my Information’ (in the upper left-hand corner of the main menu), and click on the Fetch/Validate link next to the ORCID field. This will take you to the ORCID site and allow you to create a new iD or authenticate a pre-existing iD in Editorial Manager. Please see the following video for instructions on linking an ORCID iD to your Editorial Manager account: https://www.youtube.com/watch?v=_xcclfuvtxQ 6. Please amend your list of authors on the manuscript to ensure that each author is linked to an affiliation. Authors’ affiliations should reflect the institution where the work was done (if authors moved subsequently, you can also list the new affiliation stating “current affiliation:….” as necessary). 7. Please include your full ethics statement in the ‘Methods’ section of your manuscript file. In your statement, please include the full name of the IRB or ethics committee who approved or waived your study, as well as whether or not you obtained informed written or verbal consent. If consent was waived for your study, please include this information in your statement as well. 

Reviewers' comments:

Reviewer's Responses to Questions

**Comments to the Author**

1. Is the manuscript technically sound, and do the data support the conclusions?

Reviewer #1: No

Reviewer #2: Partly

Reviewer #3: Yes

2. Has the statistical analysis been performed appropriately and rigorously? 

Reviewer #1: N/A

Reviewer #2: N/A

Reviewer #3: N/A

3. Have the authors made all data underlying the findings in their manuscript fully available?

Reviewer #1: Yes

Reviewer #2: Yes

Reviewer #3: Yes

4. Is the manuscript presented in an intelligible fashion and written in standard English?

Reviewer #1: Yes

Reviewer #2: Yes

Reviewer #3: Yes

5. Review Comments to the Author

Reviewer #1: This is an interesting article reporting on a portion of a qualitative study of older workers thinking about retirement. It does have something to offer to the literature but I think the article is wrongly targeted at this journal; the data is presented in a narrative way, which does not meet many of the criteria PLOS One has for the reporting of this kind of data. The article would be better sent to a journal that specialises in this kind of study but even then more work would need to be done on how the study was conducted, data analysed etc. At the moment we have insufficient information on:

- we have no numbers for how many managers and HR personnel were interviewed, I think the total of 92 covers them and employees and both sectors, although they are reporting mainly on the finance companies;

- how the interviews were conducted and by whom, it would be helpful to have the semi-structure

d guide in an appendix;

- how were the interviews analysed, using a software programme? Were they coded? What was the coding process?

- what was the basis for selecting the direct quotations reported in the piece- were they indicative of widely expressed views or? The ethnographic vignette for example, why this case?

The theoretical foundations for the piece and the literature review also need some attention. The article refers to meta stereotypes but it is not clear if the study was systematically looking for these or other manifestations of ageism in the interview process. It would also have been useful to give a bit more attention to the literature on embodied/internalised ageism as this is what the data seems to reveal.

The notion of ‘worn out syndrome’ sounds very similar to the concept of ‘the decline narrative’ and it would have been useful to have some discussion of this. Also a number of authors have pointed to ‘too old for’ narratives which also sound similar. We need to know if worn out syndrome is different and if so how?

The article is tantalising in telling us about the two sectors but only really reporting on one. The difference between physical and cognitive ageing and how this is embodied would be especially interesting.

The following references might be helpful:

Gullette, M. M. (2004). Aged by Culture. University of Chicago Press.

Romaioli, D., & Contarello, A. (2019). I’m Too Old for ...” Looking into a Self Sabotage Rhetoric and its Counter-narratives in an Italian Setting. Journal of Aging Studies 48, 25–32. doi: 10.1016/j.jaging.2018.12.001

Romaioli, D., & Contarello, A. (2021) Resisting ageism through lifelong learning. Mature students' counter-narratives to the construction of aging as decline. Journal of Aging Studies, 57, https://doi.org/10.1016/j.jaging.2021.100934

Spedale, S. (2018), “Deconstructing the ‘older worker’: Exploring the complexities of subject positioning at the intersection of multiple discourses”, Organization, Vol. 26 No. 1, pp. 38-54.

Van der Horst, M. (2019). “Internalised ageism and self-exclusion: Does feeling old and health pessimism make individuals want to retire early?”, Social Inclusion, Vol. 7 No. 3, pp. 27-43.

Vickerstaff, S. & van der Horst, M. (2021). The Impact of Age Stereotypes and Age Norms on Employees’ Retirement Choices: A Neglected Aspect of Research on Extended Working Lives. Frontiers in Sociology, 6:686645. doi: 10.3389/fsoc.2021.686645

Vickerstaff, S. & van der Horst, M.(2022), “Embodied ageism: “I don't know if you do get to an age where you're too old to learn” Journal of Aging Studies, Vol. 62: https://doi.org/10.1016/j.jaging.2022.101054

Reviewer #2: I enjoyed reading this manuscript. The topic is very interesting. What I was missing in the analysis is: 1. a short description of the Danish financial services industry (how big, structure, preponderant sectors) and the choice of professional positions that have been investigated. I assume that the worn out syndrome might be different in its manifestations and drivers for someone working in a bank branch compared with someone working on a trading floor, or for someone doing quantitative finance and someone doing sales in a bank. It would have been interesting to investigate how traders retire, for instance. I suspect they retire much earlier than employees of bank branches. 2. As this investigation is based on qualitative interviews, it would have been interesting to see an exploration of the positions and perceptions of those employees who have not retired: how do they perceive those retiring or approaching the retirement limit? 3. When exploring the feelings of inadequacy and/ or slowing down of the work rhythms, I missed concrete instances or examples of events or incidents that might trigger such feelings. The descriptions in the interview excerpts are rather general. For instance, fear of falling behind IT developments is mentioned but the analysis is not really developed: can we have perhaps a deeper exploration there. Overall, personally I would have structured the analysis according to professional positions (e.g., bank manager, trader, analyst, sales, etc.), educational background, and drivers of such feelings, fleshing out instances or episodes that might trigger them.

Reviewer #3: See attached file of more detailed comments. I recommend accepting with essentially copyedit revisions but possibly some removal of redundancies and brief discuss of political/economic forces/incentives driving elderly in or out of the labor market in social democratic wealthy high-tech countries.

6. PLOS authors have the option to publish the peer review history of their article (what does this mean?). If published, this will include your full peer review and any attached files.

Reviewer #1: No

Reviewer #2: **Yes: **Alex Preda

Reviewer #3: **Yes: **Philippe Bourgois

---

## [Author Response · Author response to Decision Letter 0]

7 Oct 2022

Rebuttal letter to the article Worn out syndrome

Reviewer 1

Thanks for your constructive comments, which points to some weaknesses in the manuscript that has helped us in improving it. 

- we have no numbers for how many managers and HR personnel were interviewed, I think the total of 92 covers them and employees and both sectors, although they are reporting mainly on the finance companies; - how the interviews were conducted and by whom, it would be helpful to have the semi-structured guide in an appendix;

You are right that 92 covers them, but this was not obvious in the previous version. We have inserted a paragraph on lines 241-255 which describes how many of the different kinds of interlocutors we have followed, as well as how the interviews were conducted and by whom. We have included the interview guide in an appendix. 

On lines 67-69 we describe why we are reporting mainly on the finance industry, and on lines 71-78 we now explain the Danish finance industry more in depth. 

- how were the interviews analysed, using a software programme? Were they coded? What was the coding process?

You are right that this lacked in the previous version. We have inserted a new paragraph on lines 269-276 explaining our analytic approach and process. 

- what was the basis for selecting the direct quotations reported in the piece- were they indicative of widely expressed views or? The ethnographic vignette for example, why this case?

In the new paragraph on lines 269-276 we explain that the selected quotes are indicative of widely expressed views unless noted otherwise. 

On line 286-288 we now explain why we have chosen this vignette. 

The theoretical foundations for the piece and the literature review also need some attention. The article refers to meta stereotypes but it is not clear if the study was systematically looking for these or other manifestations of ageism in the interview process. 

In the new paragraph on lines 269-276 we explain that we use an analytic inductive approach. As such, the concept of meta stereotypes and the development of worn out syndrome were results of this analytic process and not something we knew we would explore from the beginning of the fieldwork. These concepts and phenomena appeared in our data and were subsequentially confirmed in the data. 

It would also have been useful to give a bit more attention to the literature on embodied/internalised ageism as this is what the data seems to reveal.

The notion of ‘worn out syndrome’ sounds very similar to the concept of ‘the decline narrative’ and it would have been useful to have some discussion of this. Also a number of authors have pointed to ‘too old for’ narratives which also sound similar. We need to know if worn out syndrome is different and if so how?

Thanks for your suggestions. We had included some discussion on ageism in the discussion and literature review sections, but acknowledge that we could have shown how the concept relates to more literature than already included. We have therefore added on lines 192-195 and included some of the suggested literature – in particular the work of Vickerstaff and van der Horst as well as the ‘too old for’ narrative. However, we find that some of the suggested literature did not add to the argument. 

The article is tantalising in telling us about the two sectors but only really reporting on one. The difference between physical and cognitive ageing and how this is embodied would be especially interesting.

We have mentioned the production industry for transparency purposes and in order to show how there is a difference between the politization of bodily decline on the one hand and cognitive decline on the other, which we show in the discussion lines 646-657.This could be interesting to explore further in a different article. 

Reviewer 2

Many thanks for your constructive suggestions. 

1. We agree that we needed a more thorough description of the Danish finance industry and have done so on lines 71-78. While we agree that the senior working life of traders could be interesting to investigate, we have no data on this particular group. 

2. Most of the cases we have included in the paper have not retired but are approaching retirement age. As we show on lines 245-250, some of the interlocutors were managers, shop stewards and HR personnel, and from these data we have gained a glimpse into how other age groups perceive the senior employees. While this could be interesting to investigate further, this requires a different approach and a new fieldwork. 

3. We are sorry that you think we miss concrete instances. With this kind of data we cannot show causal relations between the worn out syndrome and concrete actions, but we believe that we show concrete instances throughout the paper, in which the interlocutors describe when they experience uncertainty and a feeling of being worn out. To show this more clearly, we have repeated some of these instances in the discussion lines 599-617.

Reviewer 3

Thanks for your review and your appreciative reading of the manuscript. 

We appreciate your suggestions regarding the production industry and have inserted how this could be explored in another article on line 695.

We have mentioned the production industry for transparency purposes and in order to show how there is a difference between the politization of bodily decline on the one hand and cognitive decline on the other, which we show in the discussion lines 646-657.

We sympathize with the comment on pursuing an article with a focus on the political economy of senior working life, but think that this requires a different article and scope. However, we are unfolding this more in an upcoming article stemming from the project. 

While we do think that the Nordic welfare states create a different approach to retiring or staying in the workforce than in the US or countries with resembling state constructions, we cannot say much about this from our data. In the abovementioned upcoming article, we explore the literature about this issue in more depth. 

Regarding your comment on redundancy, we agree that we have repeated the argument too much. We have therefore removed it from the lines 518-521 (in the new version, this would have been on line 587).

---

## [Decision Letter · Decision Letter 1]

12 Dec 2022

PONE-D-22-20359R1The worn-out syndrome: Uncertainties in late working life triggering retirement decisionsPLOS ONE

Dear Dr. Lassen,

Thank you for submitting your manuscript to PLOS ONE. After careful consideration, we feel that it has merit but does not fully meet PLOS ONE’s publication criteria as it currently stands. Therefore, we invite you to submit a revised version of the manuscript that addresses the points raised during the review process.

We look forward to receiving your revised manuscript.

Kind regards,

Dan-Cristian Dabija, PhD

Academic Editor

PLOS ONE

Additional Editor Comments:

Dear authors,

it seems that the reviewers still consider that the paper should be restructured and improved. Please implement accordingly their comments and suggestions.

Cristian Dabija

Reviewers' comments:

Reviewer's Responses to Questions

**Comments to the Author**

1. If the authors have adequately addressed your comments raised in a previous round of review and you feel that this manuscript is now acceptable for publication, you may indicate that here to bypass the “Comments to the Author” section, enter your conflict of interest statement in the “Confidential to Editor” section, and submit your "Accept" recommendation.

Reviewer #1: (No Response)

Reviewer #2: All comments have been addressed

Reviewer #3: All comments have been addressed

2. Is the manuscript technically sound, and do the data support the conclusions?

Reviewer #1: Partly

Reviewer #2: Yes

Reviewer #3: Yes

3. Has the statistical analysis been performed appropriately and rigorously? 

Reviewer #1: N/A

Reviewer #2: N/A

Reviewer #3: N/A

4. Have the authors made all data underlying the findings in their manuscript fully available?

Reviewer #1: No

Reviewer #2: Yes

Reviewer #3: Yes

5. Is the manuscript presented in an intelligible fashion and written in standard English?

Reviewer #1: Yes

Reviewer #2: Yes

Reviewer #3: Yes

6. Review Comments to the Author

Reviewer #1: The authors are to be commended for responding well to reviewers' comments. The revised manuscript is much better in terms of situating the study in the Danish context, explaining the research process and providing the research tools; some issues about the sample remain however. I think it is bad practice not to say how many interviews were done in each of the two sectors and although there is now more about why the finance sector is focused on it leaves open a set of questions about the productive sector. It is asserted that uncertainty was greater for white collar than blue collar but we are not presented with any data to back this up. It is counterintuitive as we would expect the manual workers to have more physical health conditions as they age and to be more uncertain about their physical ability to carry on working. I think it would make more sense to only discuss the finance sector as no data is really provided on the productive sector so the reader cannot assess any comparison being made.

We do not have any information of the interviewees pension status, there is a question asked: Why do you think you will stop working?

And ‘finance’ is provided as a prompt and a question about pension provider, I don’t know enough about pension arrangements in Denmark, perhaps everyone, whatever sector and work history, has adequate pensions but in most other places you would expect quite a difference between white collar and blue collar workers and those in finance would be likely to have very good pensions. Whilst finance may not be a major factor in retirement decisions, it is likely to be a boundary factor at least – can I afford to retire -. The fact that as state pension age has risen participation rates have increased would seem to underlie the importance of finance. In the vignette case we hear nothing about her financial situation. It may be that for this relatively privileged group of finance sector workers money isn’t much of an issue?

The framing of the piece conceptually still raises some questions. The claim of the article overall is that the authors have discovered a new framing of age in the workplace the ‘worn-out syndrome’ (WOS) but if you substitute either ‘decline narrative’ or ‘too old for’ (already existing concepts) for WOS the overall argument works just as well. To put this another way how does WOS differ from these existing ideas in the literature? Substituting WOS really only works if there is a clear argument as to its superiority over existing concepts; in fact in the discussion and conclusion there is lots of talk about decline and some of the respondents converse in these terms as well. For this reason I don’t think the revised manuscript yet reaches the PLOS requirement that “Conclusions are presented in an appropriate fashion and are supported by the data.” Relevant literature is now referenced but not directly engaged with.

Reviewer #2: The authors have reviewed the paper following the reviewers' suggestions. I take their point about the differences between the impostor syndrome and the worn out syndrome. The way the data is presented and analysed makes it difficult to tell whether this syndrome is more specific to the financial services industry compared with other service sectors or not. We know that the interviewees work in banks but there is little insight in what they do and whether the specific profile of their expertise contributes to this syndrome or not. The only more specific information we get is about an interviewee working in the pensions department of one bank and about a new IT system being introduced. But this could be public administration too, or any other service sector introducing a new IT system. I am saying this because we could have two different claims: (1) that this syndrome is particularly present in the financial services industry, or (2) that this syndrome is present in the services industry, with the financial sector being just an example. If (2), which I believe is the case based on how the data is presented here, then the authors should make this clear and argue why they selected this sector for their investigations. We get a brief sentence that it has gone through restructuring since the 2000s, but so have other sectors presumably too. I think this can be addressed with one or two sentences that make clear that the syndrome the authors discuss is not specific too or more pronounced in the financial services industry (it could be, but the authors do not have data to show this).

Reviewer #3: Wrote a clear response to each of the comments by all three reviewers. I didn't re-read the final version to check for copyedits however.

7. PLOS authors have the option to publish the peer review history of their article (what does this mean?). If published, this will include your full peer review and any attached files.

Reviewer #1: No

Reviewer #2: **Yes: **Alex Preda

Reviewer #3: **Yes: **Philippe Bourgois

---

## [Author Response · Author response to Decision Letter 1]

31 Jan 2023

Dear reviewers and editor

Many thanks for your feedback and suggestions. Below we respond to each of the points raised. 

Reviewer 1:

- I think it is bad practice not to say how many interviews were done in each of the two sectors

We agree and have corrected this on lines 229-230 and 239-243. 

- and although there is now more about why the finance sector is focused on it leaves open a set of questions about the productive sector. It is asserted that uncertainty was greater for white collar than blue collar but we are not presented with any data to back this up. It is counterintuitive as we would expect the manual workers to have more physical health conditions as they age and to be more uncertain about their physical ability to carry on working. I think it would make more sense to only discuss the finance sector as no data is really provided on the productive sector so the reader cannot assess any comparison being made.

We have gone through the manuscript, and only find the production industry mentioned in the beginning (where we briefly present the initial project and fieldwork) and in the last section of the discussion. In this latter part, we have now removed some sentences, where we used data from the production industry. We now only reflect briefly on why we find the worn-out syndrome more in the finance industry on lines 626-643. 

- We do not have any information of the interviewees pension status, there is a question asked: Why do you think you will stop working? And ‘finance’ is provided as a prompt and a question about pension provider, I don’t know enough about pension arrangements in Denmark, perhaps everyone, whatever sector and work history, has adequate pensions but in most other places you would expect quite a difference between white collar and blue collar workers and those in finance would be likely to have very good pensions. Whilst finance may not be a major factor in retirement decisions, it is likely to be a boundary factor at least – can I afford to retire -. The fact that as state pension age has risen participation rates have increased would seem to underlie the importance of finance.

Thanks for this comment. We agree that in other countries, worn-out syndrome might not play a role in retirement decisions, and have included this on lines 161-165.

- In the vignette case we hear nothing about her financial situation. It may be that for this relatively privileged group of finance sector workers money isn’t much of an issue?

The vignette case does not include the financial situation, as this did not factor in her retirement decision, and was not the reason for including the vignette.

- The framing of the piece conceptually still raises some questions. The claim of the article overall is that the authors have discovered a new framing of age in the workplace the ‘worn-out syndrome’ (WOS) but if you substitute either ‘decline narrative’ or ‘too old for’ (already existing concepts) for WOS the overall argument works just as well. To put this another way how does WOS differ from these existing ideas in the literature? Substituting WOS really only works if there is a clear argument as to its superiority over existing concepts; in fact in the discussion and conclusion there is lots of talk about decline and some of the respondents converse in these terms as well. 

We agree that worn-out syndrome relates to the concepts mentioned, but believe that it differs as well, as we now explain more clearly on lines 186-191. , The ‘decline narrative’ and ‘too old for’ rely on internalized or embodied ageism, but WOS differs in the way that the participants do not have to believe, that they are worn out themselves, but the syndrome works just as well when participants are in doubt whether their surroundings might think they are worn out. Moreover, the ‘decline narrative’ and ‘too old for’ are not standard concepts that everyone in the field relates to, and there is plenty of space to develop new, relatable concepts with other nuances. 

Reviewer #2: 

- The authors have reviewed the paper following the reviewers' suggestions. I take their point about the differences between the impostor syndrome and the worn out syndrome. The way the data is presented and analysed makes it difficult to tell whether this syndrome is more specific to the financial services industry compared with other service sectors or not. We know that the interviewees work in banks but there is little insight in what they do and whether the specific profile of their expertise contributes to this syndrome or not. The only more specific information we get is about an interviewee working in the pensions department of one bank and about a new IT system being introduced. But this could be public administration too, or any other service sector introducing a new IT system. I am saying this because we could have two different claims: (1) that this syndrome is particularly present in the financial services industry, or (2) that this syndrome is present in the services industry, with the financial sector being just an example. If (2), which I believe is the case based on how the data is presented here, then the authors should make this clear and argue why they selected this sector for their investigations. We get a brief sentence that it has gone through restructuring since the 2000s, but so have other sectors presumably too. I think this can be addressed with one or two sentences that make clear that the syndrome the authors discuss is not specific too or more pronounced in the financial services industry (it could be, but the authors do not have data to show this)

Many thanks for this suggestion. We believe that the syndrome is widespread in other service sectors, but we do not have data to show this. On lines 67-70 we now state this as well as explain how the changes in the finance industry has led to issues with mental health.

---

## [Decision Letter · Decision Letter 2]

27 Feb 2023

The worn-out syndrome: Uncertainties in late working life triggering retirement decisions

PONE-D-22-20359R2

Dear Dr. Lassen,

We’re pleased to inform you that your manuscript has been judged scientifically suitable for publication and will be formally accepted for publication once it meets all outstanding technical requirements.

Kind regards,

Dan-Cristian Dabija, PhD

Academic Editor

PLOS ONE

Additional Editor Comments (optional):

As all reviewers are pleased with the latest version of the paper, I recommend to accept it

Reviewers' comments:

Reviewer's Responses to Questions

**Comments to the Author**

1. If the authors have adequately addressed your comments raised in a previous round of review and you feel that this manuscript is now acceptable for publication, you may indicate that here to bypass the “Comments to the Author” section, enter your conflict of interest statement in the “Confidential to Editor” section, and submit your "Accept" recommendation.

Reviewer #1: All comments have been addressed

Reviewer #2: All comments have been addressed

2. Is the manuscript technically sound, and do the data support the conclusions?

Reviewer #1: Yes

Reviewer #2: Yes

3. Has the statistical analysis been performed appropriately and rigorously? 

Reviewer #1: N/A

Reviewer #2: N/A

4. Have the authors made all data underlying the findings in their manuscript fully available?

Reviewer #1: No

Reviewer #2: Yes

5. Is the manuscript presented in an intelligible fashion and written in standard English?

Reviewer #1: Yes

Reviewer #2: Yes

6. Review Comments to the Author

Reviewer #1: The authors have responded effectively to the comments from two reviewers. The article is now clearer in terms of the data set being used and the discussion is better grounded in existing literature and the Danish context. The discussion of the findings is now more nuanced and complete. Highlighting the issues of perceived cognitive decline is very useful.

Reviewer #2: I think that the authors have addressed all the questions I have raised in a satisfactory manner. They have specified their arguments in such a way that the syndrome they discuss is presented in a more nuanced and circumscribed way.

7. PLOS authors have the option to publish the peer review history of their article (what does this mean?). If published, this will include your full peer review and any attached files.

Reviewer #1: No

Reviewer #2: **Yes: **Alex Preda

---

## [Editor Report · Acceptance letter]

1 Mar 2023

PONE-D-22-20359R2 

The worn-out syndrome: Uncertainties in late working life triggering retirement decisions 

Dear Dr. Lassen:

I'm pleased to inform you that your manuscript has been deemed suitable for publication in PLOS ONE. Congratulations! Your manuscript is now with our production department. 

Kind regards, 

on behalf of

Professor Dan-Cristian Dabija 

Academic Editor

PLOS ONE